# Band-Limited Gaussian Processes:
# The Sinc Kernel

**Felipe Tobar**
Center for Mathematical Modeling
Universidad de Chile
`ftobar@dim.uchile.cl`

## Abstract

We propose a novel class of Gaussian processes (GPs) whose spectra have compact support, meaning that their sample trajectories are almost-surely band limited. As a complement to the growing literature on spectral design of covariance kernels, the core of our proposal is to model power spectral densities through a rectangular function, which results in a kernel based on the sinc function with straightforward extensions to non-centred (around zero frequency) and frequency-varying cases. In addition to its use in regression, the relationship between the sinc kernel and the classic theory is illuminated, in particular, the Shannon-Nyquist theorem is interpreted as posterior reconstruction under the proposed kernel. Additionally, we show that the sinc kernel is instrumental in two fundamental signal processing applications: first, in stereo amplitude modulation, where the non-centred sinc kernel arises naturally. Second, for band-pass filtering, where the proposed kernel allows for a Bayesian treatment that is robust to observation noise and missing data. The developed theory is complemented with illustrative graphic examples and validated experimentally using real-world data.

## 1   Introduction

### 1.1   Spectral representation and Gaussian processes

The spectral representation of time series is both meaningful and practical in a plethora of scientific domains. From seismology to medical imagining, and from astronomy to audio processing, understanding which fraction of the energy in a time series is contained on a specific frequency band is key for, e.g., detecting critical events, reconstruction, and denoising. The literature on spectral estimation [13, 24] enjoys of a long-standing reputation with proven success in real-world applications in discrete-time signal processing and related fields. For unevenly-sampled noise-corrupted observations, Bayesian approaches to spectral representation emerged in the late 1980s and early 1990s [4, 11, 8], thus reformulating spectral analysis as an inference problem which benefits from the machinery of Bayesian probability theory [12].

In parallel to the advances of spectral analysis, the interface between probability, statistics and machine learning (ML) witnessed the development of Gaussian processes (GP, [21]), a nonparametric generative model for time series with unparalleled modelling abilities and unique conjugacy properties for Bayesian inference. GPs are the *de facto* model in the ML community to learn (continuous-time) time series in the presence of unevenly-sampled observations corrupted by noise. Recent GP models rely on Bochner theorem [2], which indicates that the covariance kernel and power spectral density (PSD) of a stationary stochastic process are Fourier pairs, to construct kernels by direct parametrisation of PSDs to then express the kernel via the inverse Fourier transform. The precursor of this concept in ML is the spectral-mixture kernel (SM, [32]), which models PSDs as Gaussian

RBFs, and its multivariate extensions [29, 19]. Accordingly, spectral-based sparse GP approximations [15, 10, 5, 14] also provide improved computational efficiency.

## 1.2 Contribution and organisation

A fundamental object across the signal processing toolkit is the normalised sinc function, defined by

$$\text{sinc}(x) = \frac{\sin \pi x}{\pi x}. \tag{1}$$

Its importance stems from its role as the optimal basis for reconstruction (in the Shannon-Whittaker sense [31]) and the fact that its Fourier transform is the rectangle function, which has compact support. Our hypothesis is that the symbiosis between spectral estimation and GPs can greatly benefit from the properties of kernels inspired the sinc function, yet this has not been studied in the context of GPs. In a nutshell we propose to parametrise the PSD by a (non-centred) rectangular function, thus yielding kernels defined by a sinc function times a cosine, resembling the SM kernel [32] with the main distinction that the proposed PSD has compact, rather than infinite, support.

The next section introduces the proposed sinc kernel, its centred/non-centred/frequency-varying variants as well as its connections to sum-of-infinite-sinusoids models. Section 3 interprets posterior reconstruction using the sinc kernel from the Shannon-Nyquist perspective. Then, Sections 4 and 5 revise the role of the sinc kernel in two signal processing applications: stereo demodulation and band-pass filtering. Lastly, Section 6 validates the proposed kernel through numerical experiments with real-world signals and Section 7 presents the future research steps and main conclusions.

## 2 Compact spectral support via the sinc kernel

The Bochner theorem [2] establishes the connection between a (stationary) positive definite kernel $K$ and a density $S$ via the Fourier transform $\mathcal{F}\{\cdot\}$, that is,

$$K(t) = \mathcal{F}^{-1}\{S(\xi)\}(t), \tag{2}$$

where the function $S : \mathbb{R}^n \mapsto \mathbb{R}_+$ is Lebesgue integrable. This result allows us to design a valid positive definite function $K$ by simply choosing a positive function $S$ (a much easier task), to then *anti-Fourier transform it* according to eq. (2). This is of particular importance in GPs, where we can identify $K$ as the covariance kernel and $S$ the spectral density, therefore, the design of the GP can be performed in the spectral domain rather than the temporal/spatial one. Though temporal construction is the classical alternative, spectral-based approaches to covariance design have become popular for both scalar and vector-valued processes [32, 29, 19], and even for nonstationary [22] and nonparametric [27, 26] cases.

We next focus on GPs that are *bandlimited*, or in other words, that have a spectral density with compact support based on the sinc kernel.

### 2.1 Construction from the inverse Fourier transform of rectangular spectrum

Let us denote the rectangular function given by

$$\text{rect}(\xi) \stackrel{\text{def}}{=} \begin{cases} 1 & |\xi| < 1/2 \\ 1/2 & |\xi| = 1/2 \\ 0 & \text{elsewhere,} \end{cases} \tag{3}$$

and consider a GP with a power spectral density (PSD), denoted by $S$, given by the sum of two rectangular functions placed symmetrically[1] wrt the origin at $\xi_0$ and $-\xi_0$, with widths equal to $\Delta$ and total power equal to $\sigma^2$. We refer to this construction as the **symmetric rectangle** function with centre $\xi_0$, width $\Delta$ and power $\sigma^2$ denoted by

$$\text{simrect}_{\xi_0,\Delta,\sigma^2}(\xi) \stackrel{\text{def}}{=} \frac{\sigma^2}{2\Delta}\left(\text{rect}\left(\frac{\xi - \xi_0}{\Delta}\right) + \text{rect}\left(\frac{\xi + \xi_0}{\Delta}\right)\right), \tag{4}$$

where the denominator $2\Delta$ ensures that the function integrates $\sigma^2$ and the explicit dependence on $\xi_0, \Delta, \sigma^2$ will only be shown when required. We assume $\Delta > 0$; $\xi_0, \sigma^2 \geq 0$, and note that the rectangles are allowed to overlap if $\Delta > 2\xi_0$.

We can then calculate the kernel associated with the PSD given by $S(\xi) = \mathrm{simrect}_{\xi_0, \Delta, \sigma^2}(\xi)$ using the standard properties of the Fourier transform. In particular, we can do so by identifying the symmetric rectangle function in eq. (4) as a convolution between a (centred) rectangle and two Dirac delta functions on $\{\xi_0, -\xi_0\}$. We define this kernel as follows.

**Definition 1** (The Sinc Kernel). *The stationary covariance kernel resulting from the inverse Fourier transform of the symmetric rectangle function in eq.* (4) *given by*

$$\mathrm{SK}(t) \overset{\text{def}}{=} \sigma^2 \mathrm{sinc}(\Delta t) \cos(2\pi \xi_0 t), \tag{5}$$

*is referred to as **the sinc kernel** of frequency $\xi_0 \geq 0$, bandwidth $\Delta \geq 0$ and magnitude $\sigma^2 \geq 0$. The expression $\mathrm{sinc}(t) = \frac{\sin \pi t}{\pi t}$ is known as the the normalised sinc function, and when $\xi_0 = 0$ we refer to the above expression as the **centred sinc kernel**.*

Being positive definite by construction, the sinc kernel can be used within a GP for training, inference and prediction. Thus, we implemented a GP with the sinc kernel (henceforth **GP-sinc**) for the interpolation/extrapolation of a heart-rate time series from the MIT-BIH database [7]. Using one third of the data, training the GP-sinc (plus noise variance) was achieved by maximum likelihood, were both the BFGS [33] and Powell [20] optimisers yielded similar results. Fig. 1 shows the leant PSD and kernel alongside the periodogram for comparison, and a sample path for temporal reconstruction and forecasting. We highlight that the sinc function implemented in Python used in this optimisation was numerically stable for both optimisers and multiple initial conditioned considered.

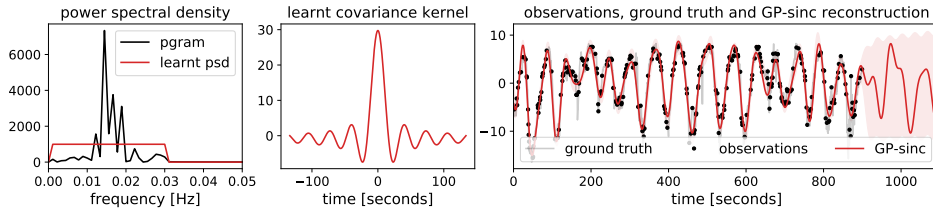

Figure 1: Implementation of the sinc kernel on a heart-rate time series. Notice that (i) the learnt kernel shares the same support as the periodogram, (ii) the error bars in the reconstruction are tight, and (iii) the harmonic content in the forecasting part is consistent with the ground truth.

## 2.2 Construction from a mixture of infinite sinusoids

Constructing kernels for GP models as a sum of infinite components is known to aid the interpretation of its hyperparameters [21]. For the sinc kernel, let us consider an infinite sum of sines and cosines with random magnitudes respectively given by $\alpha(\xi), \beta(\xi) \sim \mathcal{N}(0, \sigma^2)$ i.i.d., and frequencies between $\xi_0 - \frac{\Delta}{2}$ and $\xi_0 + \frac{\Delta}{2}$. That is,

$$f(t) = \int_{\xi_0 - \frac{\Delta}{2}}^{\xi_0 + \frac{\Delta}{2}} \alpha(\xi) \sin(2\pi \xi t) + \beta(\xi) \cos(2\pi \xi t) \mathrm{d}\xi. \tag{6}$$

The kernel corresponding to this zero-mean GP can be calculated using basic properties of the Fourier transform, trigonometric identities and the independence of the components magnitudes. This kernel is stationary and given by the sinc kernel defined in eq. (5):

$$K(t, t') = \mathbb{E}\left[f(t)f(t')\right] = \sigma^2 \mathrm{sinc}((t - t')\Delta) \cos(2\pi \xi_0 (t - t')) = \mathrm{SK}(t - t'). \tag{7}$$

The interpretation of this construction is that the paths of a GP-sinc can be understood as having frequency components that are equally present in the range between $\xi_0 - \frac{\Delta}{2}$ and $\xi_0 + \frac{\Delta}{2}$. On the contrary, frequency components outside this range have zero probability to appear in the GP-sinc sample paths. In this sense, we say that the sample trajectories of a GP with sinc kernel are *almost surely band-limited*, where the *band* is given by $\left[\xi_0 - \frac{\Delta}{2}, \xi_0 + \frac{\Delta}{2}\right]$.

## 2.3 Frequency-varying spectrum

The proposed sinc kernel only caters for PSDs that are constant in their (compact) support due to the rectangular model. We extend this construction to band-limited processes with a PSD that is a non-constant function of the frequency. This is equivalent to modelling the PSD as

$$S(\xi) = \text{simrect}_{\xi_0,\Delta,\sigma^2}(\xi)\,\Gamma(\xi), \tag{8}$$

where the symmetric rectangle gives the support to the PSD and the function $\Gamma$ controls the frequency dependency. Notice that the only relevant part of $\Gamma$ is that in the support of $\text{simrect}_{\xi_0,\Delta,\sigma^2}(\cdot)$, furthermore, we assume that $\Gamma$ is non-negative, symmetric and continuous almost everywhere (the need for this will be clear shortly).

From eq. (8), the proposed sinc kernel can be generalised for the frequency-varying case as

$$\text{GSK}(t) \stackrel{\text{def}}{=} \mathcal{F}^{-1}\left\{\text{simrect}_{\xi_0,\Delta,\sigma^2}(\xi)\,\Gamma(\xi)\right\} = \text{SK}(t) \star K_\Gamma(t), \tag{9}$$

referred to as **generalised sinc kernel**, and where $K_\Gamma(t) = \mathcal{F}^{-1}\{\Gamma(\xi)\}$ is a positive definite function due to (i) the Bochner theorem and (ii) the fact that $\Gamma(\xi)$ is symmetric and nonnegative.

The convolution in the above equation can be computed analytically only in a few cases, most notably when $K_\Gamma(t)$ is either a cosine or another sinc function, two rather limited scenarios. In the general case, we can take advantage of the compact support of the symmetric rectangle in eq. (8), and express it as a sum of $N \in \mathbb{N}$ narrower disjoint rectangles of width $\frac{\Delta}{N}$ to define an $N$-th order approximation of $\text{GSK}(t)$ through

$$\begin{aligned}
\text{GSK}(t) &= \sum_{i=1}^{N} \mathcal{F}^{-1}\left\{\text{simrect}_{\xi_0^{(i)},\frac{\Delta}{N},\sigma^2}(\xi)\,\Gamma(\xi)\right\} \\
&\approx \sum_{i=1}^{N} \Gamma\left(\xi_0^{(i)}\right) \mathcal{F}^{-1}\left\{\text{simrect}_{\xi_0^{(i)},\frac{\Delta}{N},\sigma^2}(\xi)\right\} \\
&= \text{sinc}\,\tfrac{\Delta}{N}t \sum_{i=1}^{N} \Gamma\left(\xi_0^{(i)}\right) \cos\left(2\pi\xi_0^{(i)}\right) \stackrel{\text{def}}{=} \text{GSK}_N(t),
\end{aligned} \tag{10}$$

where $\xi_0^{(i)} = \xi_0 - \Delta\frac{N+1-2i}{2N}$, and the approximation in eq. (10) follows the assumption that $\Gamma(\xi)$ can be approximated by $\Gamma\left(\xi_0^{(i)}\right)$ within $[\xi_0^{(i)} - \frac{\Delta}{2N}, \xi_0^{(i)} + \frac{\Delta}{2N}]$ supported by the following remark.

**Remark 2.** *Observe that the expression in eq. (10) can be understood as a Riemann sum using the mid-point value. Therefore, convergence of $\text{GSK}_N(t)$ to $\text{GSK}(t)$ as $N$ goes to infinite is guaranteed provided that $\Gamma(\cdot)$ is Riemman-integrable, or, equivalently, $\Gamma(\cdot)$ is continuous almost everywhere. This is a sound requirement as it is related to the existence of the inverse Fourier transform.*

## 3 Relationship to Nyquist frequency and perfect reconstruction

The Nyquist–Shannon sampling theorem specifies a sufficient condition for perfect, i.e., zero error, reconstruction of band-limited continuous-time signals using a finite number of samples [23, 17]. Since (i) GPs models are intrinsically related to reconstruction, and (ii) the proposed sinc kernel ensures band-limited trajectories almost surely, we now study the reconstruction property the GP-sinc from a classical signal processing perspective.

Let us focus on the baseband case ($\xi_0 = 0$), in which case we obtain the **centred** sinc kernel given by

$$\text{SK}(t) = \sigma^2 \text{sinc}(\Delta t). \tag{11}$$

For a centred GP-sinc, $f(t) \sim \mathcal{GP}(0, \text{sinc}_{\sigma^2,0,\Delta})$, the Nyquist frequency is given by the width of its PSD, that is, $\Delta$. The following Proposition establishes the interpretation of Nyquist perfect reconstruction from the perspective of a vanishing posterior variance for a centred GP-sinc.

**Proposition 3.** *The posterior distribution of a GP with centred sinc kernel concentrates on the Whittaker–Shannon interpolation formula [23, 31] with zero variance when the observations are noiseless and uniformly-spaced at the Nyquist frequency [17].*

*Proof.* Let us first consider $n \in \mathbb{N}$ observations taken at the Nyquist frequency with times $\mathbf{t}_n = [t_1, \ldots, t_n]$ and values $\mathbf{y}_n = [y_1, \ldots, y_n]$. With this notation, the posterior GP-sinc is given by

$$p(f(t)|\mathbf{y}_n) = \mathcal{GP}(\text{SK}(t, \mathbf{t}_n)^\top \Lambda^{-1}\mathbf{y}_n, \text{SK}(t, t') - \text{SK}(t, \mathbf{t}_n)^\top \Lambda^{-1}\text{SK}(t', \mathbf{t}_n)), \quad (12)$$

where $\Lambda = \text{SK}(\mathbf{t}_n, \mathbf{t}_n)$ is the covariance of the observations and $\text{SK}(t, \mathbf{t})$ denotes the vector of covariances with the term $\text{SK}(t, t_i) = \text{SK}(t - t_i)$ in the $i$−th entry.

A key step in the proof is to note that the covariance matrix $\Lambda$ is diagonal. This is because the difference between any two observations times, $t_i, t_j$, is a multiple of the inverse Nyquist frequency $\Delta^{-1}$, and the sinc kernel vanishes at all those multiples except for $i = j$; see eq. (11). Therefore, replacing the inverse matrix $\Lambda^{-1} = \sigma^{-2}\mathbf{I}_n$ and the centred sinc kernel in eq. (11) into eq. (12) allows us to write the posterior mean and variance (choosing $t = t'$ above) respectively as

$$\mathbb{E}[f(t)|\mathbf{y}_n] = \sum_{i=1}^n y_i \operatorname{sinc}(\Delta(t - t_i)), \qquad \mathbb{V}[f(t)|\mathbf{y}_n] = \sigma^2 \left(1 - \sum_{i=1}^n \operatorname{sinc}^2(\Delta(t - t_i))\right). \quad (13)$$

For the first part of the proof, we can just apply $\lim_{n\to\infty}$ to the posterior mean and readily identify the Shannon-Whittaker interpolation formula: a convolution between the sinc function and the observations.

To show that the posterior variance vanishes as $n \to \infty$, we proceed by showing that the Fourier transform of the sum of square sinc functions in eq. (13) converges to a Dirac delta (at zero) of unit magnitude instead, as these are equivalent statements. Denote by $\operatorname{tri}(\cdot)$ the triangular function and observe that

$$\mathcal{F}\left\{\sum_{i=1}^\infty \operatorname{sinc}^2(\Delta(t - t_i))\right\} = \mathcal{F}\left\{\operatorname{sinc}^2(\Delta t)\right\}\mathcal{F}\left\{\sum_{i=1}^\infty \delta_{t_i}\right\} \qquad \text{conv. def. \& thm.} \quad (14)$$

$$= \frac{1}{\Delta}\operatorname{tri}\left(\frac{\xi}{\Delta}\right)\Delta\sum_{i=1}^\infty \delta_{i\Delta} \qquad \text{Fourier: } \operatorname{sinc}^2(\cdot) \text{ and } \delta_{(\cdot)}$$

$$= \delta_0(\xi),$$

where the last line follows from the fact that, out of all the Dirac deltas in the summation, the only one that falls on the support of the triangular function (of width $2\Delta$) is the one at the origin $\delta_0(\xi)$. $\quad\square$

The above result opens perspectives for analysing GPs' reconstruction errors; this is needed in the GP literature. This is because a direct consequence of Proposition 3 is a quantification of the required number of observations for zero posterior variance (or reconstruction error). This is instrumental to design sparse GPs where the number of inducing variables is chosen with a sound metric in mind: proximity to the Nyquist frequency. Finally, extending the above result to the non-baseband case can be achieved through frequency modulation, the focus of the next section.

## 4 Stereo amplitude modulation with GP-sinc

We can investigate the relationship between trajectories of GPs both for non-centred—eq. (5)—and centred—eq. (11)—sinc kernels using a latent factor model. Specifically, let us consider two i.i.d. GP-sinc processes $x_1, x_2 \sim \mathcal{GP}(0, \sigma^2 \operatorname{sinc}(\Delta t))$ with centred sinc kernel and construct the factor model

$$x(t) = x_1 \cos(2\pi\xi_0 t) + x_2 \sin(2\pi\xi_0 t). \quad (15)$$

Observe that, due to independence and linearity, the process $x$ in eq. (15) is a GP with zero mean and covariance given by a non-centred sinc kernel[2]

$$K_x(t, t') = \mathbb{E}[x(t)x(t')] = \sigma^2 \operatorname{sinc}(\Delta(t - t')) \cos(2\pi\xi_0(t - t')) = \text{SK}(t - t'). \quad (16)$$

This result can also be motivated by the following decomposition of the sinc kernel:

$$\text{SK}(t - t') = \begin{bmatrix} \cos 2\pi\xi_0 t \\ \sin 2\pi\xi_0 t \end{bmatrix}^\top \begin{bmatrix} \sigma^2 \operatorname{sinc}(\Delta(t - t')) & 0 \\ 0 & \sigma^2 \operatorname{sinc}(\Delta(t - t')) \end{bmatrix} \begin{bmatrix} \cos 2\pi\xi_0 t' \\ \sin 2\pi\xi_0 t' \end{bmatrix}. \quad (17)$$

The above matrix can be interpreted as the covariance of a multioutput GP [16, 1], where the two channels $x_1, x_2$ are independent due to the block-diagonal structure. Then, the trajectories of the non-centred sinc kernel can be simulated by: (i) sampling the two channels in this MOGP, (ii) multiplying one of them by a sine and the other one by a cosine, to finally (iii) summing them together.

The outlined relationship between centred and non-centred sinc trajectories is of particular interest in stereo modulation/demodulation [18] applications from a Bayesian nonparametric perspective. This is because we can identify the two independent draws from the centred sinc kernel as lower frequency signals containing *information* (such as stereo audio, bivariate sensors, or two-subject sensors) and the deterministic higher frequency sine and cosine signals as a *carrier*. In this setting, since the paths of a GP-sinc are equal (in probability) to those of the factor model presented in eq. (15), we can consider the GP-sinc as a generative model for stereo amplitude modulation.

Recall that the very objective in stereo demodulation is to recover the latent information signals, henceforth referred to as *channels*, at the receiver's end from (possibly corrupted) observations. In this regard, the sinc kernel represents a unique contribution, since Bayesian signal recovery under noisy/missing observations is naturally handled by GP models. In simple terms, for a stereo modulated signal with carrier frequency $\xi_0$ and bandwidth $\Delta$, the posterior over channels $\{x_i\}_{i=1,2}$ wrt an observation $\mathbf{x}$ (of the modulated signal) is jointly Gaussian and given by

$$p(x_i(t)|\mathbf{x}) = \mathcal{GP}(K_{x_i,\mathbf{x}}^\top(t)\Lambda^{-1}\mathbf{x}, K_{x_i}(t-t') - K_{x_i,\mathbf{x}}^\top(t)\Lambda^{-1}K_{\mathbf{x},x_i}(t')), \tag{18}$$

where $\Lambda = \mathrm{SK}\,(\mathbf{t},\mathbf{t}) + \mathbf{I}\sigma_{\mathrm{noise}}^2$ is the covariance of the observations, $K_{x_i}(t-t')$ is the prior covariance of channel $x_i(t)$, and $K_{x_i,\mathbf{x}}(t)$ is the covariance between observations $\mathbf{x}$ and channel $x_i(t)$ given by

$$K_{x_i,\mathbf{x}}(t) = \mathbb{E}\left[x_i(t)x(\mathbf{t})\right] = \sigma^2 \mathrm{sinc}(\delta(t-\mathbf{t}))\cos(2\pi\xi_0\mathbf{t}), \tag{19}$$

where we have used the same notation as eq. (12).

Fig. 2 shows an illustrative implementation of GP-sinc demodulation, where the associated channels were recovered form non-uniform observations of a sinc-GP trajectory.

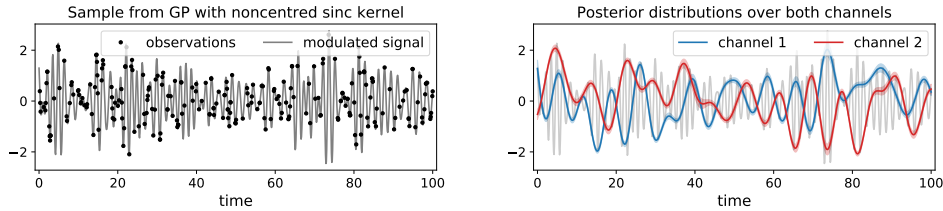

Figure 2: Demodulation using the sinc kernel. **Left:** A draw from a GP with noncentred sinc kernel (information "times" carrier). **Right:** Posterior of the stereo channels with latent modulated signal in light grey.

## 5 Bayesian band-pass filtering with GP-sinc

In signal processing, the extraction of a frequency-specific part of a signal is referred to as *band-pass filtering* [9]; accordingly, *low-pass* and *high-pass* filtering refer to extracting the low (centred around zero) and high frequency components respectively. We next show that the sinc kernel in eq. (5) has appealing features to address band-pass filtering from a Bayesian standpoint, that is, to find the posterior distribution of a frequency-specific component conditional to noisy and missing observations. For the specific low-pass filtering seeting, see [30].

We formulate the filtering setting as follows. Let us consider a signal given by the mixture

$$x(t) = x_{\mathrm{band}}(t) + x_{\mathrm{else}}(t), \tag{20}$$

where $x_{\mathrm{band}}$ and $x_{\mathrm{else}}$ correspond to independent GPs only containing energy at frequencies inside and outside the band of interest respectively. Then, we can denote the PSDs of $x(t)$ by $S(\xi)$ and those of the components by $S_{\mathrm{band}}(\xi)$ and $S_{\mathrm{else}}(\xi)$ respectively. Therefore, our assumptions of independence of the components $x_{\mathrm{band}}(t)$ and $x_{\mathrm{else}}(t)$ results on $S(\xi) = S_{\mathrm{band}}(\xi) + S_{\mathrm{else}}(\xi)$, where $S_{\mathrm{band}}(\xi)$ and $S_{\mathrm{else}}(\xi)$ have non-overlapping, or disjoint, support. An illustration of these PSDs is shown in Fig. 3.

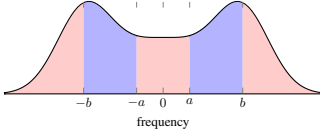

Figure 3: Illustration of PSDs in the band-pass filtering setting: The area inside the black line is the PSD of the process $x$, whereas the regions in blue and red denote the PSDs of the band component $x_\text{band}$ ($S_\text{band}$) and frequencies outside the band $x_\text{else}$ ($S_\text{else}$) respectively. Choosing $a = 0$ recovers the low-pass setting.

Notice that the above framework is sufficiently general in the sense we only require that there is a part of the signal on which we are interested, namely $x_\text{band}(t)$, and *the rest*. Critically, we have not imposed any requirements on the kernel of the complete signal $x$. Due to the joint Gaussianity of $x$ and $x_\text{band}$, the Bayesian estimate of the band-pass filtering problem, conditional to a set of observations $\mathbf{x}$, is given by a GP posterior distribution, the statistics of which will be given by the covariances of $x$ and $x_\text{band}$. Since $S_\text{band}$ can be expressed as the PSD of $x$ times the symmetric rectangle introduced in eq. (4), we can observe that the covariance of $x_\text{band}$ is given by the generalised sinc kernel presented in eq. (9) and, therefore, it can be computed via the inverse Fourier transform:

$$K_\text{band}(t) = \mathcal{F}^{-1}\left\{S_\text{band}(\xi)\right\} = \mathcal{F}^{-1}\left\{S(\xi)\operatorname{simrect}_{a,b}(\xi)\right\} = K(t) \star \operatorname{SK}(t), \qquad (21)$$

where $K(t)$ denotes the covariance kernel of $x$. Recall that this expression can be computed relying on the Riemann-sum approximation for the convolution presented in Sec. 2.3. Then, the marginal covariance of $x_\text{band}$ can be computed from the assumption of independence[3]

$$\mathbb{V}\left[x(t), x_\text{band}(t')\right] = \mathbb{E}\left[x_\text{band}(t)x_\text{band}(t')\right] + \underbrace{\mathbb{E}\left[x_\text{else}(t)x_\text{band}(t')\right]}_{0} = K_\text{band}(t - t'). \qquad (22)$$

In realistic filtering scenarios we only have access to noisy observations $\mathbf{y} = [y_1, \ldots, y_n]$ at times $\mathbf{t} = [t_1, \ldots, t_n]$. Assuming a white and Gaussian observation noise with variance $\sigma_\text{noise}^2$ and independent from $x$, the posterior of $x_\text{band}$ is given by

$$p(x_\text{band}(t)|\mathbf{y}) = \mathcal{GP}(K_\text{band}(t - \mathbf{t})\Lambda^{-1}\mathbf{y}, K_\text{band}(t - t') - K_\text{band}(t - \mathbf{t})^\top \Lambda^{-1} K_\text{band}(t' - \mathbf{t})), \qquad (23)$$

where $\Lambda = K(\mathbf{t}, \mathbf{t}) + \sigma_\text{noise}^2 \mathbf{I}$ is the covariance of the observations and recall that $K_\text{band}(t) = K(t) \star \operatorname{SK}(t)$ from eq. (21)

To conclude this section, notice that the proposed sinc-kernel-based Bayesian approach to band-pass filtering is consistent with the classical practice. In fact, if no statistical knowledge of the process were available for $x$, we can simply assume that the process is uncorrelated and the observations are noiseless. This is equivalent to setting $K(t) = \delta_0(t)$, $\Lambda = \mathbf{I}$, and $K_\text{band}(t) = \operatorname{SK}(t)$, therefore, we recover the "brick-wall" [18] filter:

$$\hat{x}_\text{band}(t) = \sum_{i=1}^{n} \operatorname{sinc}\Delta(t - t_i)\cos 2\pi\xi_0(t - t_i)y_i. \qquad (24)$$

## 6 Experiments

We validated the ability of the proposed sinc kernel to address, in probabilistic terms, the problems of (i) band-limited reconstruction, (ii) demodulation and (iii) band-pass filtering using real-world data. All examples included unevenly-sampled observations.

### 6.1 Reconstruction of a band-limited audio signal

We considered an audio recording from the TIMIT repository [6]. The signal, originally sampled at 16kHz, was low-pass filtered using a brick-wall filter at 750Hz. We focused on the reconstruction setting using only 200 (out of 1000) observations with added Gaussian noise of standard deviation equal to a 10% of that of the audio signal. Fig. 4 shows the PSDs of the true and GP-sinc reconstructed signals (mean and sample trajectories), where it can be seen that the proposed reconstruction follows faithfully the spectral content of the original signal, i.e., it does not introduce unwanted frequency components.

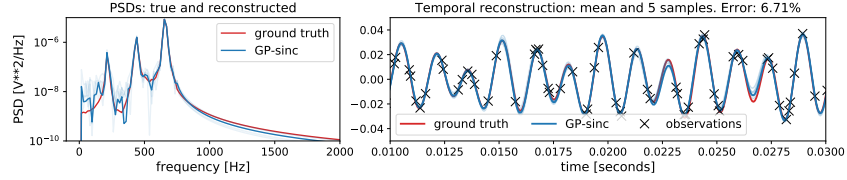

Figure 4: Band-limited reconstruction using GP-sinc: PSDs (left) and temporal reconstruction (right)

For comparison, we also reconstructed the band-limited audio signal with a GP with spectral mixture kernel (GP-SM) and a cubic spline. Fig. 5 shows the PSDs of the complete signal in red and those of the reconstructions in blue for GP-SM (left) and the cubic spline (right). Notice how the proposed GP-sinc (Fig. 4, left) outperformed GP-SM and the spline due to its rectangular PSD, which allows frequencies with high and zero energies to be arbitrarily close, unlike that of GP-SM that does not allow for a PSD with sharp decay.

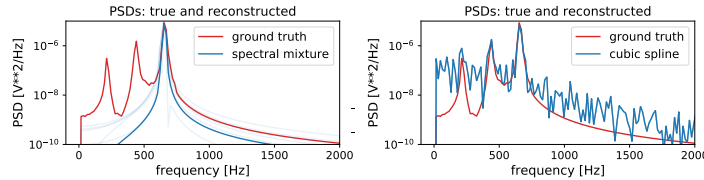

Figure 5: Reconstruction of a band-limited audio signal using GP-SM (left) and cubic spline (right). Ground truth PSD is shown in red and reconstructions in blue.

## 6.2 Demodulation of two heart-rate signals

We considered two heart-rate signals from the MIT-BIH Database [7], upsampled from 2Hz to 10Hz, corresponding to two different subjects, which can thus be understood as statistically independent. We then composed a stereo modulated signal using carrier of frequency 2Hz (most of the power of the heart-rate signals is contained below 1Hz), and used a subset of 1200 (out of 9000) observations samples with added noise of standard deviation equal to a 20% of that of the modulated signal. Fig. 6 shows the 35-run 10-90 percentiles for the reconstruction error for both channels versus the average sampling frequency (recall that these are unevenly-sampled series), and the temporal reconstruction for sampling frequency equal to 0.167. Notice how the reconstruction of the channels reaches a plateau for frequencies greater than 0.06, suggesting that oversampling does not improve performance as suggested by Proposition 3. The discrepancy in reconstruction error stems from the richer spectrum of channel 1.

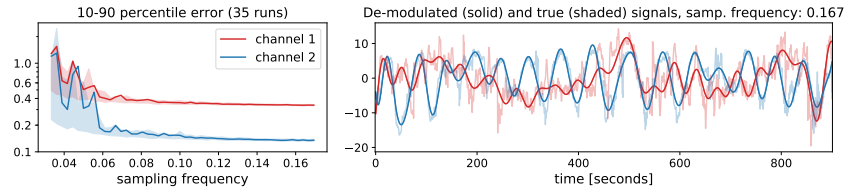

Figure 6: Heart-rate demodulation using GP-sinc: error (left) and reconstruction (right).

## 6.3 Band-pass filtering of $CO_2$ concentration

We implemented GP-sinc for extracting the 1-year periodicity component of the well-known Mauna-Loa monthly $CO_2$ concentration series. We used 200 (out of 727) observations, that is, an average sampling rate of $0.275[\text{month}^{-1}] \approx 3.3[\text{year}^{-1}]$, which is above the Nyquist frequency for the desired component. Fig. 7 shows both the unfiltered and the GP-sinc filtered PSDs (left), and the latent signal, observation and band-pass version using GP-sinc with $\xi_0 = [\text{year}^{-1}]$ and $\Delta = 0.1$.

Notice that, as desired, the GP-sinc band-pass filter was able to recover the yearly component from non-uniformly acquired observations.

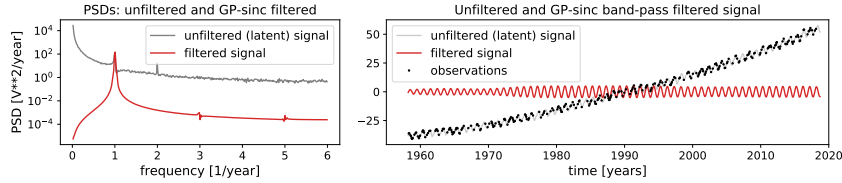

Figure 7: Bandpass filtering of Mauna-Loa monthly $CO_2$ concentration using GP-sinc.

### 6.4 Generalised sinc kernel and Nyquist-based sparse implementation

Lastly, we implemented the generalised sinc kernel (GSK) in eq. (9), i.e., a sinc mixture, using a sparse approximation where inducing locations are chosen according to the Nyquist frequency—see Sec. 3. We trained a GP with the GSK kernel, using the heart rate signal from the MIT-BIH database where we simulated regions of missing data. Fig. 8 shows the PSD at the left (components in colours and GSK in red), the resulting sum-of-sincs kernel at the centre, and the time series (ground truth, observations, and reconstruction) at the right. Notice from the right plot that though $N = 600$ observations were considered (black dots), only $M = 54$ inducing locations (blue crosses) were needed since they are chosen based on the extension of the support of the (trained) PSD (Sec. 3).

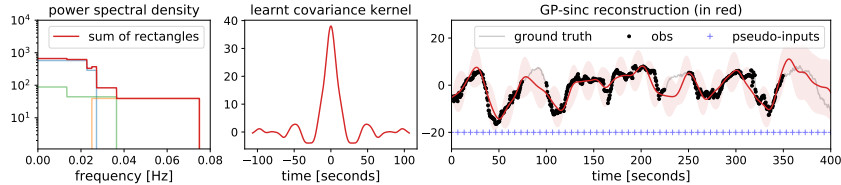

Figure 8: Implementation of generalised sinc kernel (sum of sincs) and Nyquist-based sparse approximation using a heart-rate signal. From left to right: PSDs (components in colour and sum in red), resulting GSK kernel and heart-rate signal.

## 7 Discussion

We have proposed a novel stationary covariance kernel for Gaussian processes (GP), named the sinc kernel, that generates trajectories with band-limited spectrum. This has been achieved by parametrising the GP's power spectral density as a rectangular function, and then applying the inverse Fourier transform. In addition to its use on GP training and prediction, the properties of the proposed kernel have been illuminated in the light of the classical spectral representation framework. This allowed us to interpret the role of the sinc kernel on infinite mixtures of sinusoids, Nyquist reconstruction, stereo amplitude modulation and band-pass filtering. From theoretical, illustrative and experimental standpoints, we have validated both the novelty of the proposed approach as well as its consistency with the mature literature in spectral estimation. Future research lines include exploiting the features of the sinc kernel for sparse interdomain GP approximations [14] and spectral estimation [25], understanding the error reconstruction rates for the general kernels following the results of Section 3, and comparing general kernels via a mixture of sinc kernels as suggested in Section 2.3.

### Acknowledgments

This work was funded by the projects Conicyt-PIA #AFB170001 Center for Mathematical Modeling and Fondecyt-Iniciación #11171165.

## Footnotes

[1]We consider PSDs that are symmetric wrt the origin since we focus on the real-valued GPs. Nevertheless, the presented theory can be readily extended to non-symmetric PSDs that would give rise to complex-valued covariances and thus complex-valued GP trajectories [3, 28]

[2]This follows directly from the identity $\cos(\alpha_1 - \alpha_2) = \cos(\alpha_1)\cos(\alpha_2) + \sin(\alpha_1)\sin(\alpha_2)$ choosing $\alpha_i = 2\pi\xi_0 t_i$ for $i = 1, 2$

[3]We can extend this model and assume that $x_\text{band}$ and $x_\text{else}$ are correlated, this is direct from the MOGP literature that designs covariance functions between GPs.

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
