[Reviews · NeurIPS 2019]

Reviewer 1



NOVELTY & SIGNIFICANCE This paper presents an application of GP to a wide range of signal processing problems via the spectral construction of a sinc kernel that results in band-limited GP which carries a lot of task-specific insights. Although the paper mostly applies the spectral representation of GP, I find the application non-trivial. Furthermore, it also generates an interesting theoretical result that connects GP to existing theories in signal reconstruction, which also opens a new perspective on a principled metric to choosing inducing variables for sparse GP approximation. I consider this a novel contribution. TECHNICAL SOUNDNESS Much of the derivation is a direct consequence of applying the spectral representation of GP which makes sense to me. The drawn relationship between the band-limited GP and Nyquist frequency and perfect reconstruction is less clear to me -- I can follow all equations but I am not familiar with Nyquist frequency and perfect reconstruction to make a detailed check here -- perhaps the authors should consider giving a quick recap on such literature to improve readability. The discussion paragraph on principled method to select inducing variables from lines 155 to 160 also does not read well to me -- it is not exactly clear what the authors mean by proximity to the Nyquist frequency -- I suppose this is some sort of distance between the Nyquist frequency and the predictive mean induced by the sinc kernel but this should be defined mathematically and definitely needs more elaboration. EXPERIMENT The experiments are well presented. However, comparison with existing GP methods on the same problem benchmark is not reported -- this is necessary to demonstrate that by accounting for the band-limited correlation structure of data, the performance improves. CLARITY The paper is mostly well-written. Certain parts of the paper still lack intuition, which I have highlighted above. REVIEW SUMMARY This is an interesting paper with a well-motivated application. The result also connects GP to existing theory in signal processing which draws further insights on how a sparse GP might be designed to optimize its choice of the inducing variables. The experiment is well-presented but needs more comparison with existing baselines to demonstrate the practical significance of the proposed kernel. --- Post-rebuttal Feedback: Thank you for your detailed response, which has sufficiently addressed my questions. I have increased my rating of your work. There is one concern that is raised during our discussion: The authors should've demonstrated and verified empirically that as the no. of evenly spaced observations increases, the variance (approximately) reduces and the posterior become more centered around the ground-truth. That is, the effect of putting the inducing points at the Nyquist frequency needs to be demonstrated more thoroughly to strengthen the practical significance of the paper. Please do consider these suggestions while revising your paper.

Reviewer 2



In this paper, the authors have proposed the sinc kernel. By transferring the time series analysis setup to the spectral domain they parameterise the power spectral density (PSD) of the signal by a (non-centred) rectangular function, which results in a kernel defined by the sinc function multiplied by a cosine. They continue their analysis by making a connection between the Shannon-Nyquist frequency and posterior reconstruction under the sinc kernel. Finally, they propose extensions to mixtures of sinc kernels for dealing with frequency varying cases. They authors evaluate GP models with the sinc kernel under real data and have demonstrated the properties of the proposed in the cases of stereo-amplitude modulation, signal reconstruction and band-pass filtering. I am really torn about this paper. In general, the paper is well written and nicely presented. In the first pass through it I really enjoyed reading it, while I also found the results quite interesting. However, the more I started paying attention to the details the less satisfied I was. Let me explain: First of all the whole paper is about the proposition of a new kernel, the sinc kernel. However, even that is not entirely novel since such kernels did exist in the past. For instance the proposed sinc kernel is very similar to the SM kernel from [28]. The only difference is that here the authors replace the RBF kernel with the sinc function. I understand that the sinc function has more desirable properties than the RBF in the spectral domain, i.e., band-limited support. These properties further justify the results on band-pass filtering and stereo modulation. However, I do not believe that this is enough to warrant a NeurIPS publication. Leaving the novelty issue aside, I am more unsatisfied/disappointed from the fact that the authors have only mentioned a couple of interesting ideas but they have left the actual demonstration and evaluation of them for future work. More specifically: 1) Perhaps the most interesting part of the paper is Proposition 3, were the authors directly make the connection between the Shannon-Nyquist frequency and posterior reconstruction of a GP with the sinc kernel. This is a great result but it had only been evaluated under the demodulation experiment where the authors saw that higher sample frequency does not improve the performance. There is a direct hint in the paper in the end of Section 3 on how such a property could be proven very useful in the sparse GP setting to pick the number of inducing inputs, having a proper metric in mind. However, they left it for future work. 2) The whole section about inference with frequency-varying spectrum (Section 2.3) is a very nice idea with useful properties, however, there is no experimental evaluation of it, and it is left for future work too. I believe that in a technical paper where people propose a method to handle a particular problem they should also demonstrate the efficacy of it. Otherwise I do not see the reason of why it should be mentioned in the first place. My final comment has to do with the band-pass filtering using the GP-sinc. In the end of Section 5, the authors make the assumption regarding an uncorrelated process and noiseless observations in order to derive their final filter form. Are these assumptions necessary? I can see how assuming an uncorrelated process where $K(t)=\detla_0(t)$ is convenient for the convolution in the time domain, but what happens in more interesting cases? Why do you also need to assume a zero noise? Overall, I have to say that I am not negative about the paper but I am sad to say that I really do not see how it can get in at the current state. ------- Post rebuttal I have carefully read both the authors' response and the reviewers' comments. I appreciate the authors' effort and I believe that the additional experiments are a step to the right direction. I still have my concerns though. The extra experiment with the demonstration of the sparse GPs and the mixture of GP sinc looks a promising start but it is not studied thoroughly. Let me explain: - The number of inducing points is selected automatically based on the optimal frequency. This is great but we should have seen the effect of removing or adding extra points. The authors should have definitely provided log predictive densities scores in order to support the claim that we will not see any further improvement by adding more inducing points. Also, regarding the "perfect, i.e. zero-variance posterior reconstruction" when the inducing points are chosen based on the Nyquist frequency, from what I understand from the figure it seems like far fewer inducing point are used. Looks like we only use the inducing points at the locations where the uncertainty band narrows down. I am not very convinced from this plot. - The results for the mixture of sincs are good. However, I would also like to see first the kernel on a toy problem where we would already know the ground truth band in order assess the ability to recover the correct filter. Overall, what I want to say is that I really like the paper but I find it incomplete. I am increasing my score though to 5 to acknowledge the authors' effort during the rebuttal period.

Reviewer 3



The authors develop a kernel which resembles the spectral-mixture kernel, however, provides compact spectral support. It was demonstrated that specifying a GP prior with compact spectral support is valuable in the signal processing applications of demodulation and band-pass filtering. An extension of the developed kernel was also made which allows for non-constant power spectral density throughout its support. Lastly, an interesting theoretical contribution which can quantify the number of observations needed to have zero reconstruction error for their proposed kernel. I thought this was a thorough and very well written paper with clear connections made to applications. I would argue that some of the content may be more relevant to a signal processing community, however, I expect that much of the results will also be of interest to the NeurIPS community as well. I was pleased that the paper was written in an approachable manner for readers outside the signal processing community. In the experiments it wasn't clear whether your frequency varying spectrum approach was used in the studies or not. I had hoped it would be present in some of the studies even if only to demonstrate a sensible parameterization (or a priori selection) of Gamma on a real-world problem. In the empirical studies section there was no comparison made with traditional signal processing techniques. This made it difficult to assess how your proposed GP approach fits into the state-of-the-art in the field. --------------- (after rebuttal) After carefully reading the author rebuttal and the reviews I will maintain my original rating, and my recommendation for acceptance of this work. I thank the authors for providing the additional experiments in the rebuttal, however, I agree with R2 that some improvements can be made to these studies to help justify the results of the paper. I may also suggest that the authors consider a little demonstration of proposition 3 (i.e. a GP with the centred sinc kernel concentrated with zero variance using observations that are noiseless and uniformly-spaced at the Nyquist frequency). This simple demonstration could be a nice visual to augment your result and assist the reader in comprehension.

[Author Response · NeurIPS 2019]

**Author response for NeurIPS submission 6935 (Band-Limited Gaussian Processes: The Sinc Kernel)**

We are very grateful to all three Reviewers for their valuable suggestions and encouraging comments, these have
undoubtedly improved our work. We next address the concerns raised by the Reviewers in 5 parts. The following
content (in extended form) and the required code will be part of our final submission if applicable.

**I. Novelty and comparison to other covariance kernels.** We agree that, as pointed out by Reviewer 2, the proposed
Sinc kernel can be regarded as a Spectral Mixture (SM) kernel, where the RBF is replaced by a Sinc function. However,
notice that most kernels are slight modifications of one another when parametrised in the Fourier domain: SM comes
from a non-centred RBF, Square Exponential comes from a centred RBF, Cosine comes from a Dirac, $\nu$-Matérn comes
from the filter $\frac{1}{(1+\xi^2)^{\nu+1/2}}$, Exponential comes from a Student's $t$-density and, lastly, Sinc comes from a Rectangle.
Therefore, we emphasise our contribution is not the Sinc kernel *per se*, but rather a study about its unique features and
how they can enable modern GP-based approaches to Signal Processing. With the ever-growing applications of GPs,
we believe that understanding the properties of new kernels is of interest for researchers and practitioners alike.

**II. Impact on sparse GP design.** Reviewers 1 and 2 recognised the impact of our analysis towards the design of sparse
GPs and they recommended us to further explore this connection. To do so, let us first recall that implementing a
sparse GP requires (i) the quantity of inducing points $M$ (usually chosen by hand) and their locations (usually jointly
optimised with the hyperparameters). If the proposed Sinc kernel is considered, we can rely on the Shannon-Nyquist
theorem to parametrise the locations of the inducing inputs as an evenly-spaced grid at (twice) the Nyquist frequency,
i.e., the largest frequency in the spectral support. This procedure has two key advantages for sparse GP regression: first,
both the number and locations of inducing variables are determined by the Nyquist frequency (the hyperparameter $\Delta$
of the kernel) and they need not be chosen or optimised, thus simplifying the optimisation stage. Second, we have
theoretical guarantees for perfect, i.e, zero-variance, posterior reconstruction, a property that has been little explored in
the GP community. Specifically, if the range of our data is $L$ and the Nyquist frequency is $\Delta$, we can choose $M = 2L\Delta$
inducing points evenly spaced across the range of our data, which makes the training cost $\mathcal{O}(NL^2\Delta^2)$. Notice that
this rationale paves the way to quantifying the intuition that the number of inducing points should depend of both the
extension of our datapoints ($L$) and the data's frequency content ($\Delta$).

**III. Additional experiments.** The reviewers highlighted key aspects in our manuscript that were discussed but lacked
experimental validation. Next, we present two sets of experiments addressing the comparison of GP-sinc against
spectral mixture and classic interpolation relationship (see Part III.A), and implementation of the generalised Sinc
kernel, i.e., a Sinc mixture, using the previous concept of sparse approximation (explained in Part III.B).

**III.A Comparison against spectral**
**mixture and classic interpolation.**
We considered the reconstruction of
a band-limited audio signal (form
TIMIT) using 20% unevenly-sampled

observations. The plots show the spectral densities of the complete signal in red and those of the reconstructions in
blue for the proposed GP-sinc (left), GP spectral mixture (centre) and a cubic spline (right). The proposed GP-sinc
outperformed the benchmarks due to the rectangular PSD function, which allows frequencies with high and zero
energies to be arbitrarilty close.

**III.B Mixture of Sincs and sparse**
**implementation.** We trained a GP
with the generalised Sinc kernel
(GSK), i.e., a Sinc mixture, using a
heart rate signal with unobserved re-
gions. Recall that the GSK provides
a compact-support and frequency-

varying spectrum, therefore, we trained a sparse version of GSK using the rationale described in Part II. The plots show
the PSD at the left (components in colours and mixture in red), the kernel at the centre, and the time series (ground
truth, observations, and reconstruction) at the right. Notice from the right plot that though $N = 600$ observations were
considered (black dots), only $M = 54$ inducing locations (blue crosses) were needed following the discussion in Part II.

**IV. Uncorrelated process and noiseless observations (Sec. 5).** Reviewer 2 asked for the necessity of these as-
sumptions. We clarify that we were only interpreting classical Signal Processing approaches into our setting, where
no time correlation or observation model is assumed. In such case, we showed that our model collapses to the
Whittaker–Shannon interpolation. However, without this assumption our model still holds through eq. (23).

**V. The Nyquist frequency (NF).** We agree with Reviewer 1 in their recommendation to include a brief introduction to
NF and the concept of perfect reconstruction. Many thanks for this suggestion.

[Meta-Review · NeurIPS 2019]

There was a thorough discussion about the paper and the rebuttal. Based on the arguments of the reviewers, I recommend acceptance for the paper. Reviewer 2 has raised a couple of valid concerns during the discussion period and I would strongly ask the authors to address these concerns in the final version of the paper. Please, also follow up on the comments by all the reviewers. I am sure this will be a much more solid contribution if those comments are followed in the final version. Use appendices to address these concerns, if necessary.